# Phenotypic Alterations in Erythroid Nucleated Cells of Spleen and Bone Marrow in Acute Hypoxia

**DOI:** 10.3390/cells12242810

**Published:** 2023-12-10

**Authors:** Kirill Nazarov, Roman Perik-Zavodskii, Olga Perik-Zavodskaia, Saleh Alrhmoun, Marina Volynets, Julia Shevchenko, Sergey Sennikov

**Affiliations:** 1Laboratory of Molecular Immunology, Federal State Budgetary Scientific Institution “Research Institute of Fundamental and Clinical Immunology”, 630099 Novosibirsk, Russia; kirill.lacrimator@mail.ru (K.N.); zavodskii.1448@gmail.com (R.P.-Z.); perik.zavodskaia@gmail.com (O.P.-Z.); saleh.alrhmoun1@gmail.com (S.A.); mrsmarinavolynets@gmail.com (M.V.); shevchenkoja2023@yandex.ru (J.S.); 2Department of Natural Sciences, Novosibirsk State University, 630090 Novosibirsk, Russia

**Keywords:** acute hypoxia, erythroid cells, C-lectin receptors, chemokines

## Abstract

Hypoxia leads to metabolic changes at the cellular, tissue, and organismal levels. The molecular mechanisms for controlling physiological changes during hypoxia have not yet been fully studied. Erythroid cells are essential for adjusting the rate of erythropoiesis and can influence the development and differentiation of immune cells under normal and pathological conditions. We simulated high-altitude hypoxia conditions for mice and assessed the content of erythroid nucleated cells in the spleen and bone marrow under the existing microenvironment. For a pure population of CD71+ erythroid cells, we assessed the production of cytokines and the expression of genes that regulate the immune response. Our findings show changes in the cellular composition of the bone marrow and spleen during hypoxia, as well as changes in the composition of the erythroid cell subpopulations during acute hypoxic exposure in the form of a decrease in orthochromatophilic erythroid cells that are ready for rapid enucleation and the accumulation of their precursors. Cytokine production normally differs only between organs; this effect persists during hypoxia. In the bone marrow, during hypoxia, genes of the C-lectin pathway are activated. Thus, hypoxia triggers the activation of various adaptive and compensatory mechanisms in order to limit inflammatory processes and modify metabolism.

## 1. Introduction

Erythropoiesis involves several stages of proliferation and differentiation, ranging from hematopoietic stem cells to specialized erythroid precursors that later develop into mature erythrocytes [1]. This complex process is intricately intertwined with iron metabolism and is regulated by cytokines, including the main glycoprotein hormone erythropoietin (EPO) [2]. The high rate of erythrocyte production is balanced by the rate of erythrocyte turnover in the spleen and liver, maintaining the number of erythrocytes in circulation and optimizing oxygen delivery without impairing the rheological properties of blood [3]. The rate of erythropoiesis may increase significantly above the baseline in response to hypoxic stress, which occurs when there is insufficient oxygen supply to the tissues [4]. Increasing EPO levels and promoting erythroid cell differentiation can compensate for blood loss and hypoxia [5]. 

Hypoxia is a unique physiological state that has been comprehensively studied for a long time. The systemic responses and physiological changes occurring during hypoxia are well known. However, despite decades of extensive research, the molecular mechanisms governing these responses and changes are still not fully understood [6]. It has been established that hypoxia can lead to metabolic rearrangements at the cellular, tissue, and organismal levels [7]. When exposed to hypoxia conditions, modeled in vitro, and various factors (SCF, Epo, Bmp4, and others), hematopoietic cells demonstrated increased proliferation of erythroid burst-forming units [8].

It is well known that erythroid cells can affect the development and differentiation of other hematopoietic cells, including the entire immune response in general, both under normal physiological and pathological conditions, and this occurs through the synthesis and secretion of cytokines [9,10,11] and various immunoregulatory properties [12,13]. Nevertheless, it is necessary to clarify how hypoxia impacts the erythron structure and the behavior of erythroid cells, in particular their ability to synthesize cytokines. Our study focuses on analyzing the phenotypic alterations in cells of the erythroid sprout of murine bone marrow and spleen under acute hypoxia, as well as at normoxia as a control. Also, we investigated the production of immunoactive proteins and immunoregulatory genes by erythroid cells in vitro. We expect that our findings will contribute to unveiling the fundamental mechanisms of auto- and paracrine regulation of hematopoiesis by erythroid cells under the conditions of increased demand for erythrocytes in the organism.

## 2. Materials and Methods

### 2.1. Mice

We received mice from the vivarium of the Institute of Cytology and Genetics (Novosibirsk) for experiments in which we simulated high-altitude hypoxia. The control and experimental groups of mice consisted of six mice. These were males, first-generation hybrids ♂ F1 CBA × C57Bl6, 3–5 months of age.

We housed mice under normal vivarium conditions, with unlimited access to water and food while maintaining a natural dark/light cycle. The conditions for keeping mice, conducting the experiment, humane treatment, and anesthesia when removing the animal from the experiment were approved by the local committee of the RIFCI.

### 2.2. Hypoxia Model

Acute hypoxia was simulated by placing the mice in a hyperbaric chamber where a negative pressure of approximately ~−46 kPa was maintained for 16 h. Such a pressure corresponds to a rise to an altitude of over 4500 m above sea level. Thus, our model corresponds to high-altitude hypoxia. After the exposure was over, the mice were transferred to the standard conditions of a conventional vivarium. Organ retrieval was performed 3 days after the start of the experiment. For control, we used mice of the same sex and age, which were kept under the normal atmospheric pressure conditions corresponding to the geographical location. The mice were sacrificed by dislocation of the cervical vertebrae under general anesthesia with isoflurane using the RWD Life Science anesthesia machine.

### 2.3. Cell Isolation

Femurs and spleens were harvested from mice aseptically. Bone marrow cells were obtained by washing the marrow canal with PBS. Splenocytes were obtained by homogenizing the whole spleen in a glass homogenizer. Next, the splenocytes were centrifuged in a density gradient Ficoll–Urografin (ro = 1.119 g/cm^3^) (Ficoll: PanEco, Moscow, Russia; Urografin: BAYER SCHERING PHARMA, AG, Leverkusen, Germany) for 30 min at 322 RCF (Elmi, Rīga, Latvia) and washed twice with PBS to remove RBC and granulocytes. After counting, the cells were isolated via magnetic separation.

### 2.4. Isolation of Erythroid Cells

We isolated erythroid cells from the bone marrow and spleen using positive magnetic selection for the CD 71 marker. To accomplish this, we used biotinylated antibodies against CD71 (#113803, Biolegend, San Diego, CA, USA) and streptavidin-coated magnetic beads (#480015, Biolegend, San Diego, CA, USA) in accordance with the manufacturer’s protocols (MojoSort™ Streptavidin Nanobeads Column Protocol—Positive Selection). We counted the number and viability of the resulting CD-71+ using a Countess 3 automatic cell counter (Thermo Fisher Scientific, Waltham, MA, USA) according to the manufacturer’s protocols using trypan blue (Paneco, Moscow, Russia). The cell viability after isolation was at least 95%.

### 2.5. Flow Cytometry Analysis of Erythroid Cells

Mononuclear cells of the spleen and bone marrow were stained with fluorochrome-labeled monoclonal antibodies. The cells were pre-incubated with TruStain FcX™ PLUS (anti-mouse CD16/32) Antibody (Biolegend, San Diego, CA, USA, cat# 156604) reagent in an amount of 0.25 µg/million cells in a volume of 100 µL for 10 min at a temperature of 2–8 °C to block non-specific binding. The erythroid cells in the spleen mononuclear cell population were defined as cells negative for leukocyte markers and lymphoid cell markers (CD45-CD11b-CD3-Ly-6A/E (Sca-1)-CD45R/B220-, then linearly negative cells) and positive for TER-119, CD71, and CD44 markers. We used monoclonal antibodies manufactured by Biolegend (USA): Pacific Blue™ anti-mouse TER-119/Erythroid Cells Antibody (cat# 116232, 0.5 µL/million cells), PE anti-mouse/human CD44 Antibody (cat#, 103024, 5 µL/M cells) APC anti-mouse CD71 Antibody (cat# 113820, 0.625 µL/M cells), FITC anti-mouse CD45 Antibody (cat# 103108, 0.5 µL/M cells) FITC anti-mouse/human CD11b Antibody (cat# 101206, 0.5 µL/million cells), FITC anti-mouse Ly-6A/E (Sca-1) Antibody (cat# 108106, 2 µL/million cells) Alexa Fluor^®^ 488 anti-mouse/human CD45R/B220 Antibody (cat# 1103225, 4 µL/million cells), FITC anti-mouse CD3 Antibody (cat# 100204, 2 µL/million cells). Immediately prior to cytometry, 7AAD (7-AAD Viability Staining Solution, cat# 420404, 5 µL/million cells) was added to all samples. We performed flow cytometry on an Attune NxT flow cytometer (Thermo Fisher Scientific, Waltham, MA, USA) [14]. The stages of erythroid cells were identified using the Kaluza Analysis software version 2.2.1. The results are presented as the mean ± standard error (SE) of the mean. An unpaired *t*-test with Welch correction was used to determine statistically significant differences between the erythroid cells under normal and hypoxic conditions.

### 2.6. Culturing of Erythroid Cells and Harvesting of Conditioning Medium

CD 71-positive erythroid cells were cultured in serum-free X-VIVO 15 medium at a concentration of 1 million/mL supplemented with insulin–transferrin ×1 (×100 ITS, Biolot, St. Petersburg, Russia) for 24 h to obtain culture medium for subsequent measurement of cytokine concentrations. We added BSA to a final concentration of 0.5% to the resulting conditioned medium and stored it at −80 °C until analysis.

### 2.7. Analysis of Cytokine Production in Conditioned Medium of Erythroid Cells

We analyzed cytokine levels in a conditioned medium of CD 71-positive mouse erythroid cells (*n* = 5–6) using the 23-Plex Bio-Plex Pro kit (#M60009RDPD, BioRad, Hercules, CA, USA) according to the manufacturer’s recommendations. For analysis, 50 μL of conditioned medium was used, and the analysis was performed in duplicate. Samples were analyzed using a specialized BioPlex Reader 200 workstation. The resulting fluorescent signal recording data were processed using Bio-Plex Manager software (version 6.1) in accordance with five-parameter curve fitting and converted to picograms per milliliter.

### 2.8. Transformation of Multicolour Flow Cytometry Data

We manually gated the Ter-119+ cell population using the usual software for Attune™NxT Software version 3.2.1 (ThermoFisher, Waltham, MA, USA)and exported “.fcs” files. We then converted the “.fcs” files to “.csv” files using a custom Python 3.13 code via Jupyter Notebooks6.5.4. We then performed *arcsinh*-transformation with the automatically selected cofactors of the flow cytometry data contained in the “.csv” files to automatically split cells into negative and positive for all the markers. We then performed simultaneous batch correction and data normalization by *fdaNorm* and exported the corrected “.csv” files as “.fcs” files. Both *arcsinh*-transformation and *fdaNorm*-normalization were carried out using the R scripts published by Melsen et al. [15]. For dimensionality reduction and clustering, we utilized HSNE (hierarchical stochastic neighbor embedding) that constructs a hierarchical representation of the entire dataset, maintaining the non-linear high-dimensional relationships between cells that can be gradually explored from a broader overview to the detailed single-cell level [16]. HSNE was applied in the Cytosplore app [17], an integrated single-cell analysis framework that enables dynamic exploration of the hierarchy through a two-dimensional scatter plot in which cells are arranged according to the similarity in the expression of all markers at the same time. FSC, SSC, CD44, CD45, CD71, and TER-119 were used for dimensionality reduction and clustering. We then identified clusters of erythroid cells that represent different stages of erythroid differentiation. For this purpose, we relied on two factors. First, we had prior knowledge that CD71 expression is highest at the proerythroblast stage and continuously decreases until it disappears at the reticulocyte stage, and Ter-119 is present on all mouse erythroid cells. Second, we assumed that adjacent stages of differentiation would cluster close to each other. We then exported the frequencies of the cells per cluster and visualized them in GraphPad Prism 9.4.1.

### 2.9. Data Analysis: Cytokine Secretion Data

We log_2_-transformed our Bio-Plex cytokine data using Pandas 2.0.3. A heatmap was created via Bioinfokit 2.1.0 [18]. Multiple *t*-tests with FDR correction were performed in GraphPad Prism 9.4.1. to analyze the differential cytokine production. FC > 1.8 or FC < −1.8 and q-values < 0.05 were considered statistically significant.

### 2.10. Total RNA Extraction

We isolated total RNA from 500,000 CD71-enriched erythroid cells with the Total RNA Purification Plus Kit (Norgen Biotek, Thorold, ON, Canada). We measured the concentration and quality of the total RNA in each sample on a Qubit 4 (Thermo Fisher Scientific, Waltham, MA, USA). We froze the total RNA at −80 °C until the gene expression analysis.

### 2.11. Study of Gene Expression Using the NanoString nCounter SPRINT Profiler Analytical System

nCounter technology is based on direct digital target detection using fluorescent barcodes. To analyze total RNA samples, we used the nCounter Mouse Immunology v1 panel (561 immune-related genes, 15 housekeeping genes, 6 positive controls, and 8 negative controls). We used 100 ng of total RNA from each sample in a volume of 5–14 μL to hybridize with 3 μL of nCounter Reporter probes, 0–7 μL of DEPC-treated water, 11 μL of hybridization buffer, and 5 μL of nCounter Capture probes (total reaction volume = 33 µL).

The total number of target molecules was determined using the nCounter digital analyzer. nSolver 4.0 software was used for normalization and quality control using the synthetic positive controls included in the panel. To remove non-expressing genes, we performed background thresholding on the normalized data. The background level was determined as the average of the least positive controls, and genes that were below the background level in at least one sample were removed.

### 2.12. Differential Gene Expression and Cytokine Secretion

We performed log2 transformation on the gene expression (number of targets) and cytokine secretion data (concentration). We performed a differential analysis of gene expression and a differential analysis of cytokine secretion using a multiple *t*-test. We considered q values < 0.01–0.001 for gene expression analysis and q values < 0.05 for cytokine secretion analysis to be significant. The analysis and plotting of volcano plots were performed in GraphPad Prism 9.4.1.681.

## 3. Results

### 3.1. Acute Hypoxia Leads to Redistribution of Erythroid Cells in Hematopoietic Organs

We analyzed mouse erythroid cell number and phenotype in bone marrow and spleen in normal conditions and after acute hypoxia. The gating strategy is summarized in Figure 1. The phenotype and maturation stage of CD71+ erythroid cells were determined by the expression of CD 71, Ter119, and FSC [19,20,21]. We isolated a population of Ter119highCD71-positive cells and further categorized them according to forward scattering (FSC) parameters into different stages of terminal differentiation. As a result, we identified basophilic erythroid cells as Ter119highCD71posFSChigh cells, polychromatophilic erythroid cells as Ter119highCD71posFSCmiddle cells, and orthochromatophilic erythroid cells/reticulocytes as Ter119highCD71posFSClow cells. FSC allowed the Ter119highCD71posFSClow orthochromatophilic erythroid cell/reticulocyte population to be clearly separated into two parts, thus ensuring the differentiation of orthochromatophilic erythroid cells from reticulocytes.

The total number of erythroid cells from all stages of differentiation was defined as the parent population of the living cells of the organ.

The analysis demonstrated that acute hypoxia in both the bone marrow and spleen of mice resulted in a decrease in the content of erythroid nucleated cells compared to normal conditions (Figure 2). This fact was also confirmed visually by flow cytometry data, with a dramatic decrease observed in the population of Ter119highCD71pos cells under hypoxia (Figure 3B).

The analysis of the subpopulation composition revealed distinct segregation of spleen and bone marrow cells into two clearly defined populations: lineage-negative stromal cells (CD45-/CD11b-/CD3-/Ly-6A/E (Sca-1)-/CD45R/B220-) and lineage-positive lymphoid cells (CD45+/CD11b+/CD3-Ly-6A/E (Sca-1)-CD45R/B220-). Hypoxia in the spleen and bone marrow alters cellular composition by increasing the content of lineage-positive lymphoid cells, predominantly CD45-positive, and decreasing the content of lineage-negative non-lymphoid cells, predominantly CD45-negative (Figure 3A). Figure 4 shows the content of CD45-positive and CD45-negative cells in the spleen and bone marrow under normal and hypoxic conditions.

Erythroid cells were found to be predominantly present in a population of lineage-negative stromal non-lymphoid cells (CD45-/CD11b-/CD3-/Ly-6A/E (Sca-1)-/CD45R/B220-). However, the cell population positive for CD45+/CD11b+/CD3+/Ly-6A/E (Sca-1)+/CD45R/B220+ lymphoid cell lineage markers also contained erythroid cells with the CD71+Ter-119+ phenotype. Therefore, we will use the term CD 45-positive/CD45-negative erythroid cells.

As with our previously studied normal bone marrow and spleen [22], we applied HSNE dimensionality reduction to Arcsinh-transformed and fdaNorm-normalized erythroid cell flow cytometry parameters (FSC, SSC, CD44, CD45, CD71, and Ter-119) and also showed the presence of clusters corresponding to CD45-positive and CD45-negative erythroblasts and identified successive stages of erythroid cell differentiation (Figure 5), including proerythroblasts. CD45-positive erythroid cells have more pronounced side scattering, which reflects the granularity of the cell compared to CD45-negative erythroblasts at the same stage of differentiation.

The analysis of flow cytometry data demonstrated that the content of CD45-positive erythroid cells in bone marrow was equal to 5.82 ± 1.15% (mean ± SE) under normal conditions and 6.63 ± 1.67% (mean ± SE) under hypoxia. In the spleen, under normal conditions, the content of CD45-positive erythroid cells was 21.35 ± 3.76% (mean ± SE), and under hypoxia, it was equal to 27.46 ± 6.10% (mean ± SE). Significant differences were found between the content of the erythroid cells of both subpopulations in the bone marrow and spleen under either physical condition (Figure 6).

Next, we analyzed the content of basophilic, polychromatophilic, and orthochromatophilic erythroid cells and reticulocytes in each organ under normal and hypoxia conditions (Figure 7). Staging analysis of erythroid cells demonstrated significant changes in the erythroid structure. Orthochromatophilic erythroid cells were identified as the predominant population in both organs under normal and hypoxic conditions. Under hypoxia, both in the spleen and bone marrow, the content of orthochromatophilic erythroid cells was significantly decreased, while the content of polychromatophilic erythroid cells was significantly increased. The content of basophilic erythroid cells under hypoxia was found to be significantly increased only in the bone marrow. The content of reticulocytes did not change significantly (Table 1).

The next step was to determine the ratio of CD45-positive and CD45-negative erythroid cells for each stage of differentiation. CD45-positive cells were found to predominate in the population of basophilic erythroid cells in the spleen under normal conditions, with their number significantly decreasing under hypoxia. The bone marrow is characterized by the predominance of CD45-negative cells in the population of basophilic erythroid cells, with their number also significantly decreasing under hypoxia. The content of CD45-positive and CD45-negative basophilic erythroid cells was found to differ significantly between organs in both normal and hypoxic conditions (Table 2, Figure 8).

Polychromatophilic erythroid cells are characterized by a predominance of CD45-negative subpopulations in both organs under normal and hypoxia conditions. At the same time, the content of CD45-negative and CD45-positive erythroid cells was unchanged during the transition from norm to hypoxia, and statistically significant differences were shown only between organs (Table 3, Figure 9).

Orthochromatophilic erythroid cells are characterized by a predominance of CD45-negative subpopulations in both organs under normal and hypoxia conditions. The content of CD45-negative and CD45-positive erythroid cells in the bone marrow was found to be unchanged during the transition from norm to hypoxia. The spleen under hypoxia is characterized by a significant difference in the content of CD45-negative and CD45-positive erythroid cells compared to the norm. For hypoxia, the differences between organs were demonstrated (CD45-negative and CD45-positive erythroid cells), while for normal conditions, only CD45-negative erythroid cells were characterized (Table 4, Figure 10).

### 3.2. Analysis of CD71 Cytokine Secretion by Erythroid Cells under Acute Hypoxia

We investigated the production of 23-cytokines by CD71+ erythroid cells from bone marrow and spleen under normal and acute hypoxia conditions. Next, we analyzed the CD71+ cytokines differentially secreted by erythroid cells (Figure 11). The highest cytokine production was observed for CCL11, IL12p70, CCL2, CCL3, and CCL4. Medium levels of cytokines were found for CCL-5, TNF-alpha, IL-9, GM-CSF, CXCL-1, IL-13, IL-2, IFN-gamma, IL-12p70, and IL-10. The lowest levels of cytokines were found to be characteristic of IL-3, IL-1beta, IL-17a, IL-4, IL-6, G-CSF, IL-1a, and IL-5. The comparison of bone marrow and spleen in normoxia revealed that erythroid cells of bone marrow produced significantly more of the IL-1a, IL-1b, IL-2, IL-4, IL-9, IL-12p70, IL-13, G-CSF, TNFa, IFNg cytokines, and CCL2, CCL4, and CXCL1 chemokines (Figure 12c). The production of cytokines in each of the organs did not change during hypoxia, except for CCL-2. Its production by the erythroid cells of the spleen was significantly decreased during hypoxia (Figure 12d).

### 3.3. Acute Hypoxia Results in Up-Regulation of C-Lectin Receptor Signaling Pathway Genes in Murine Bone Marrow Erythroid Cells

We then performed differential gene expression analysis between the post-acute hypoxia adult bone marrow and normal adult bone marrow erythroid cells. We used a stringent fold change criterion (log2(FC) > 2/log2(FC) < −2) to avoid any gene expression differences due to the different proportions of erythroid cells at the different stages of differentiation present. *Clec4e*, *Clec5a*, *Irak3*, *Il6*, and *Tnf* genes were up-regulated, and the *Cxcl12* gene was down-regulated in post-acute hypoxia bone murine marrow erythroid cells compared to normal bone marrow murine erythroid cells (Figure 13). Gene Set Enrichment Analysis of the up-regulated genes revealed that this set of genes is enriched in the “C-lectin type receptor signaling pathway” KEGG term (Figure 14, Table 5).

## 4. Discussion

Hypoxia occurs when the oxygen content and pressure in a cell become lower than normal. The causes of hypoxia include hypoxemia (low oxygen content and pressure in the blood), impaired oxygen delivery, and impaired oxygen uptake/utilization by cells. Hypoxia can be compensated by a variety of mechanisms, both at the level of the whole organism and individual cells [23]. The classical physiological response to systemic hypoxia is increased erythrocyte production [24]. Our work analyzes the functional properties of erythroid cells under oxygen deficiency, with these cells being the main cell population required to compensate for this condition.

Efficient stress-induced erythropoiesis requires the population of immature progenitors to be augmented prior to the initiation of differentiation. This process requires the translation of proliferative signals, such as Wnt2b, 8a, and GDF15, into metabolic changes that support proliferation and prevent changes in chromatin structure associated with the activation of the erythroid gene program [25]. The compensatory mechanisms during acute hypoxia involve various changes in the erythroid cell population in mouse hematopoietic organs to realize compensatory mechanisms. In mice, the spleen remains an important erythropoietic organ even in adulthood after embryonic development. Under erythroid stress conditions, such as low oxygen pressure or anemia, the mouse spleen is used as a peripheral organ for enhanced erythropoiesis [26] and as a major organ of extramedullary erythropoiesis during stress in mice [27]. However, our study has revealed a decrease in erythroid cells both in the spleen and bone marrow, which may be due to their accelerated maturation and release into the bloodstream to compensate for tissue hypoxia. A similar effect has been described in patients with COVID-19: erythroid cells were found in their peripheral blood, with this phenomenon directly correlating with the severity of the infection course, while COVID-19 is characterized primarily by tissue hypoxia, especially in the moderate and severe course [28].

Under oxygen deficiency conditions, the distribution of erythroid cells by stages of terminal differentiation is found to differ significantly from the norm. Thus, the content of orthochromatophilic erythroid cells in the bone marrow decreases with a simultaneous increase in polychromatophilic and basophilic erythroid cells. In the spleen, the population of basophilic erythroid cells remains unchanged. It is possible for the population of orthochromatophilic erythroid cells to function under normal conditions as a regulator of erythrocyte output into the bloodstream, thereby maintaining the optimal number of erythrocytes without deterioration of blood viscosity parameters. Under hypoxia conditions, the population of orthochromatophilic erythroid cells should rapidly supply precursor cells for rapid enucleation. The activity of the orthochromatophilic population should be maintained by the activation of cells in precursor stages. Thus, the proliferative potential of erythroid cells is regulated by the accumulation of a sufficient number of precursor cells to rapidly replenish the pool of orthochromatophilic erythroid cells.

The hypoxia condition results in the appearance of a population of proerythroblasts in the spleen and bone marrow, according to Arcsinh-transformation data. Under steady-state conditions, up to 60% of proerythroblasts in the spleen of mice are thought to die due to apoptosis. Also, a decrease in the rate of apoptosis of erythroid precursors is assumed to contribute to the increase in erythrocyte production that occurs under erythropoietic stress [29]. Active migration of reticulocytes from hematopoietic organs into the peripheral bloodstream is also observed under hypoxic exposure. The analysis of markers of human erythroid cells under hypoxia in vitro allowed us to suggest that hypoxia promotes the formation of proerythroblasts (ProE) but may not affect the transition ProE→BasoE→OrthoE→PolyE [30].

Our work demonstrates that a population of CD45-positive erythroid cells is stably present in the bone marrow and spleen. However, the proportion of CD45-positive erythroid cells in the spleen is significantly higher than in the bone marrow. At the same time, the content of CD45-positive erythroid cells in mouse bone marrow has been found to coincide with the same index for human bone marrow erythroid cells [13]. CD45 expression is characteristic of the early stages of erythroid cell differentiation [12]. However, similar cells with the CD45+CD71+Ter119+ phenotype are often found in tumors as independent cells or as a product of trans-differentiation of other cells [31]. The main function of such cells is immunosuppression and suppression of effector T-cell functions [32,33]. Thus, a significant number of regulatory T cells and immunosuppressive myeloid subpopulations, including M2 macrophages and type 2 dendritic cells with low HLA-DR expression, were found in tumor regions with high hypoxia [34]. The proliferation of T-cells is suppressed during a long stay in high-altitude conditions [35]. We investigated the distribution of CD45-negative and CD45-positive erythroid cells within each stage of differentiation under normal and hypoxic conditions. Since the content of CD45-positive erythroid cells was observed to remain stable in each organ during hypoxia, we hypothesized that each stage might involve changes in subpopulation composition in terms of the presence of the CD45 marker. The predominance of CD45-positive erythroid cells in the spleen, specifically basophilic erythroid cells, aligns with the data indicating the preferential expression of this marker on earlier-stage erythroid cells. Additionally, their quantity diminishes under hypoxic conditions. Similarly, a decrease in the content of CD45-positive basophilic erythroid cells in the bone marrow was observed. This effect can be explained by the accelerated transition of earlier stages to more mature ones. The population of polychromatophilic CD45-positive erythroid cells remains stable both in the bone marrow and in the spleen. Orthochromatophilic erythroid cells in the spleen are characterized by a significant increase in CD45-positive erythroid cells during hypoxia. The population of CD45-positive erythroid cells of the bone marrow remains practically unchanged, participating only in the maintenance of more mature forms of erythroid cells, but the response of CD45-positive erythroid cells of the spleen is more pronounced. We believe that one of the essential compensatory mechanisms is the preservation of the population of CD45-positive erythroid cells until the later stage of terminal differentiation, including orthochromatophilic erythroid cells. The dramatic increase in bone marrow iron demand during hypoxia is associated with the stimulation of erythropoiesis [36]. Most of the iron used for normal erythropoiesis is recycled from phagocytized erythrocytes [37]. Thus, it can be assumed that the accumulation of erythroblasts of later stages, close to enucleation, in hematopoietic organs may be associated with increased phagocytosis of erythrocytes to maintain iron homeostasis and increase the number of oxygen-carrying erythrocytes.

CD45-positive erythroid cells exhibit more pronounced side scattering, reflecting the granularity of the cell [38] compared to CD45-negative erythroid cells of the same stage of differentiation. The granular structure of erythroid cells may result from the accumulation of glycogen granules, especially in orthochromatophilic erythroid cells. The accumulation of glycogen may indicate a preparatory mechanism for providing glycolysis substrates to nucleus-deprived erythrocytes. In mature erythrocytes devoid of functional mitochondria, energy metabolism is based on glycolysis. Moreover, the stimulation of the pentose phosphate pathway by glucose also leads to the production of NADPH, the main coenzyme required for the replenishment of glutathione-based antioxidant defense [39]. The glycogen stores appearing at the late stages of differentiation are a switch signal for the synthesis of specialized proteins that allow mature erythrocytes to be fully functional [40]. Thus, during terminal differentiation, CD45-positive erythroid cells form a cluster of cells capable of implementing both the erythroid program before the formation of a functional erythrocyte and the synthesis of immunosuppressive proteins.

The analysis of cytokine production has demonstrated chemokines to be significant mediators of erythroid cells and their production to be practically unchanged under hypoxic exposure but different across the organs of hematopoiesis. To date, cell migration, especially of leukocytes, has been studied most comprehensively as a function of the chemokine network [41]. Chemokines are crucial for the movement, migratory behavior, cellular interactions, and positioning of T cells, B cells, and DCs required to generate primary and secondary adaptive immune responses [42]. In some cases, chemokines can drive the migration of groups of cells (called collective migration) or stimulate cell adhesion, causing cells to stop moving. It is believed that cells move down, rather than up, along the chemokine concentration gradients, i.e., away from the source of chemokines, a process referred to as chemorepulsion or chemofugetaxis [43]. Erythroid cells of the bone marrow and spleen are known not only to secrete chemokines but also to produce cytokines necessary for regulating immune cell functions [44]. Hypoxia is accompanied by a pronounced change in the cellular composition of hematopoietic organs: the stroma of the organs is filled with lymphoid cells, and the predominantly non-lymphoid cells become a smaller population. In light of the above, it is assumed that the active secretion of chemokines by erythroid cells plays a role in preserving the cellular composition of bone marrow and spleen by controlling the movement of lymphoid cells, activating erythroid precursor cells, and ensuring the timely release of cells into circulation. Erythroid cells in the spleen and bone marrow have a similar cytokine and chemokine secretion profile but differ in quantity. Our assumption is that this is due to the shared functional characteristics of the erythroid cells in these organs. However, in the spleen, a short-term deposition and rapid renewal of the composition of cellular elements occur. In addition, the bone marrow is the most important storage reservoir for the vital functions of hematopoiesis support functioning, requiring, therefore, greater protection against changes in external factors. CCL2 is the only chemokine whose production by the erythroid cells of the spleen is reduced during hypoxic exposure. When binding to its CCR2 receptors, CCL2 (MCP-1) activates monocytes and other immune cells that contribute to inflammation. It directs leukocyte infiltration and also affects T-cell proliferation and immune function [45]. Our data suggest that reduced CCL2 production under hypoxia may limit the migration and inflammatory functions of macrophages, thereby impairing antigen presentation and further realization of the immune response, which is in good agreement with earlier data [46,47,48]. However, in the presence of hypoxia, medullary erythroid cells produce roughly the same level of CCL2 as they do under normal conditions. This phenomenon favors the retention of macrophages within the bone marrow to facilitate erythropoiesis, according to the erythroblastic islet concept [49]. Stress erythropoiesis is part of a coordinated inflammatory response that allows bone marrow hematopoiesis to concentrate on the production of immune effector cells, with erythroid homeostasis maintained by extramedullary stress erythropoiesis [50]. During acute hypoxia, genes in the C-lectin receptor signaling pathway are activated. C-type lectin receptors recognize “non-self” pathogen-associated molecular patterns (PAMPs) or modified endogenous “damage-associated” antigens/damage-associated molecular patterns (DAMPs) derived from dead cells or endogenous fragments of “altered self” [51]. CLEC5A is positively associated with immune infiltration [52]. Mincle (CLEC4E) acts as a prototypical activating CLR upon recognition of glycolipids in the cell wall of some fungal and bacterial pathogens and leads to the induction of several cytokines and chemokines, including TNF-α, macrophage inflammatory protein 2 (MIP-2; CXCL2), a chemokine derived from keratinocytes (KC; CXCL1), and IL-6 [53]. The C-type lectin receptor CLEC4E and the Toll-like receptor TLR4, expressed by host cells, are two of the first lines of defense when encountering pathogens [54]. IRAK3 (also known as IRAK-M, as it is primarily found in monocytes) suppresses TLR signaling, resulting in decreased production of pro-inflammatory cytokines [55]. Thus, gene activation in bone marrow erythroid cells during acute hypoxia is aimed at limiting the inflammatory response to environmental pathogens. This is important since we see an influx of lymphoid elements into the bone marrow; accordingly, any “self” or “foreign” stimulus can lead to extensive inflammation and death of the body.

To conclude, the model of acute high-altitude hypoxia demonstrated changes in the structure of hematopoietic organs actively filled with lymphoid cells and in the structure of the erythroid cell population in the foci of normal and extramedullary erythropoiesis. During hypoxia, it is typical for erythroid cells to rapidly decrease the population of orthochromatophilic erythroid cells, which are primed for rapid enucleation. The cells of previous stages of differentiation actively compensate for this process in conjunction with accelerated maturation. Hypoxia triggers the activation of various adaptive and compensatory mechanisms with the goal of restricting inflammatory processes and modifying metabolism.

## 5. Conclusions

Thus, under pathological conditions of hematopoiesis, erythroid cells perform not only the functions of compensating for erythropoiesis but also actively participate in the implementation of immunological functions and the protection of foci of hematopoiesis from damage. Modern methods today make it possible to increase the technological level of research in the field of studying the role of erythroid cells and fill gaps in understanding the dependence of the functions of erythroid cells on the degree of their differentiation and molecular immunological context. All this makes the study of immunomodulatory properties, the transcriptional profile of erythroid cells, and their influence on various populations of immunocompetent cells in normal and pathological hematopoiesis both relevant and timely.

## Figures and Tables

**Figure 1 cells-12-02810-f001:**
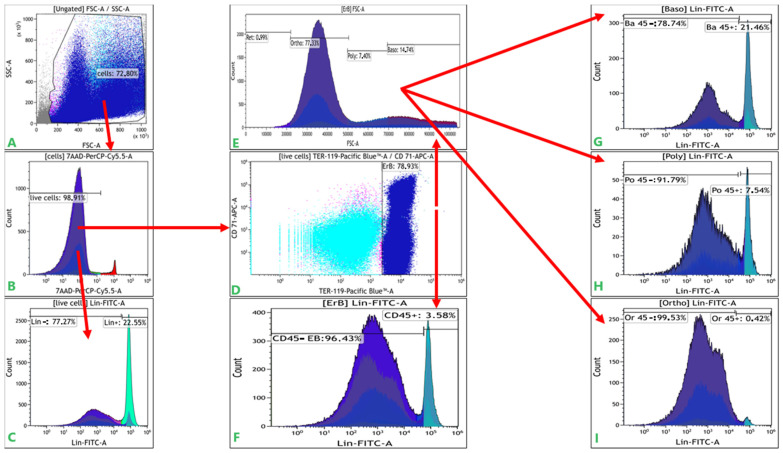
The gating strategy of flow cytometry data. (**A**)—dot-plot graph FSC vs. SSC for all cells of an organ; (**B**)—fluorescence histogram of the 7-AAD dye channel for detection of living cells; (**C**)—fluorescence histogram of lineage markers from a live cell gate to determine line-negative and line-positive populations; (**D**)—dot-plot graph Ter-119 vs. CD 71 from a live cell gate to determine erythroid cells (ErB); (**E**)—histogram of cell distribution according to the FSC parameter from the gate of erythroid cells to determine the stages of differentiation of erythroid cells (basophilic (Baso), polychromatophilic (Poly), orthochromatophilic (Ortho); (**F**)—fluorescence histogram of lineage markers from the erythroid cells gate to determine CD45-positive and CD45-negative erythroid cells; (**G**–**I**)—fluorescence histogram of lineage markers from the basophilic (Baso), polychromatophilic (Poly), orthochromatophilic (Ortho erythroid cells gate to determine CD45-positive and CD45-negative (basophilic (Baso), polychromatophilic (Poly), orthochromatophilic (Ortho) erythroid cells, respectively.

**Figure 2 cells-12-02810-f002:**
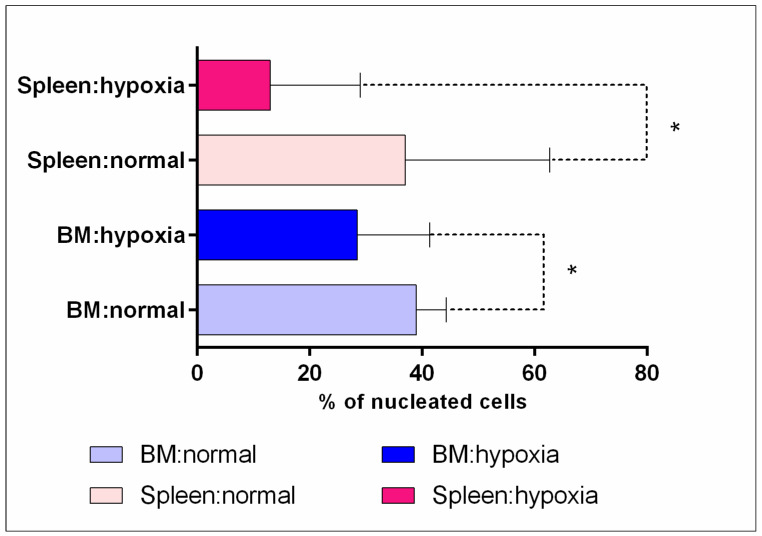
The content (relative number) of erythroid nucleated cells in the spleen and bone marrow under normal conditions and acute hypoxia. *—the differences are statistically significant (*p* < 0.05) between the content of spleen and bone marrow cells in normal conditions and in hypoxia.

**Figure 3 cells-12-02810-f003:**
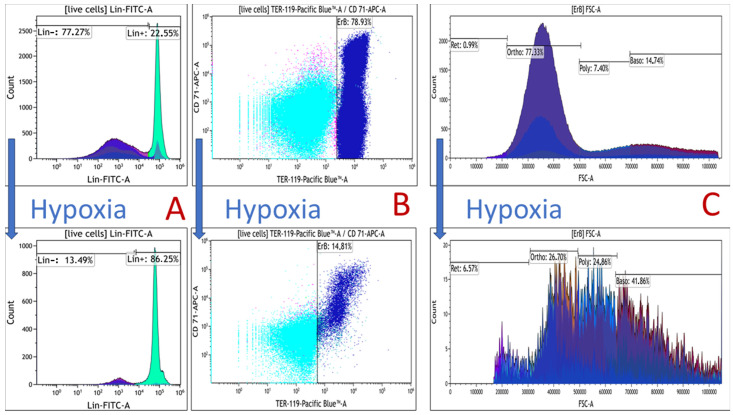
Histograms illustrate the qualitative changes in the distribution of spleen/bone marrow cells under normal and hypoxia conditions. (**A**) the change in the ratio of lineage-positive to lineage-negative cells in the spleen; (**B**) the change in the number of erythroid cells available; and (**C**) the change in the distribution of the stages of terminal differentiation of erythroid cells.

**Figure 4 cells-12-02810-f004:**
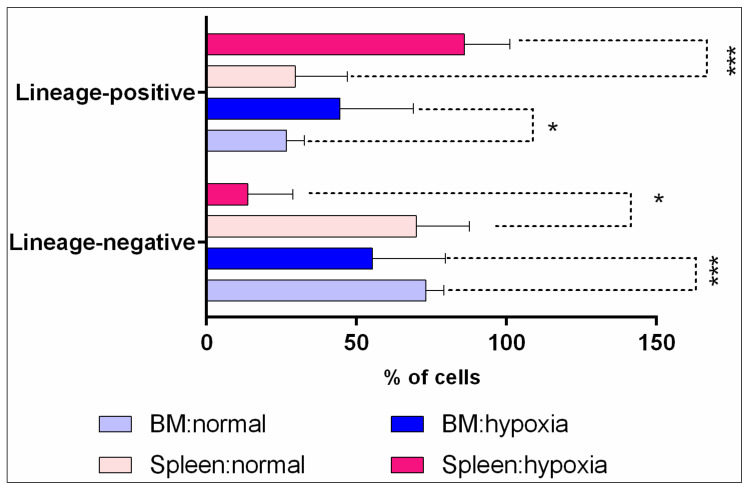
The content (relative number) of lineage-positive lymphoid cells (predominantly CD45-positive) and lineage-negative non-lymphoid cells (predominantly CD45-negative) in the spleen and bone marrow of mice in normal and hypoxic conditions. *—the differences are statistically significant (*p* < 0.05) between the content of spleen and bone marrow cells in normal conditions and in hypoxia. ***—the differences are statistically significant (*p* < 0.01) between the content of spleen and bone marrow cells in normal conditions and in hypoxia.

**Figure 5 cells-12-02810-f005:**
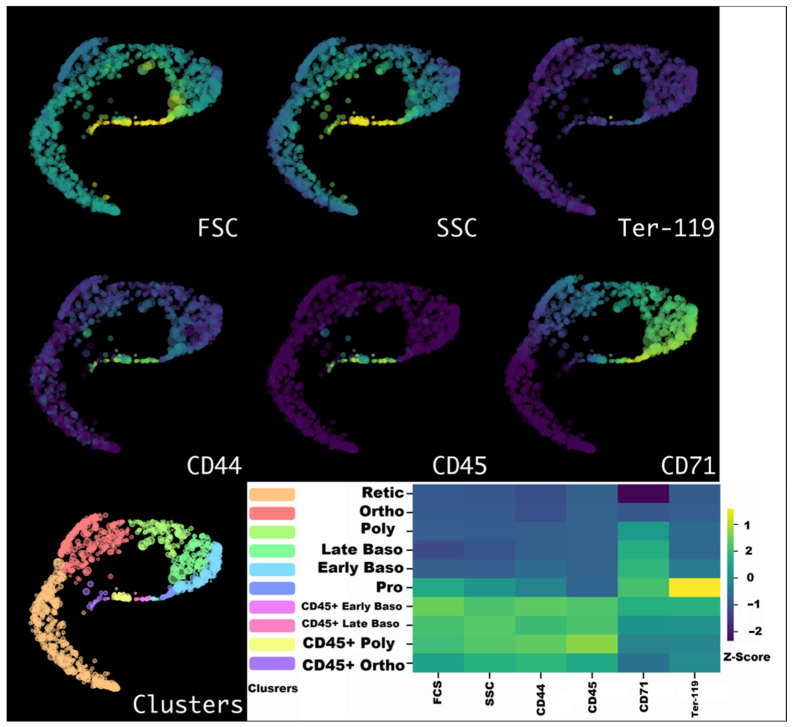
HSNE plots of erythroid cells from the bone marrow overlaid with marker expression—purple represents the absence of the marker expression, whereas yellow represents the maximum of the marker expression. Clusters are color-labeled in accordance with the heatmap. The heatmap shows the Z-score transformed mean marker expression values. Yellow represents the highest standardized expression, and purple represents the lowest standardized expression. Pro stands for proerythroblast; EarlyBaso stands for early basophilic erythroblast; LateBaso stands for late basophilic erythroblast; Poly stands for polychromatophilic erythroblast; and Ortho stands for orthochromatophilic erythroblast. Retic stands for reticulocyte. CD45+ depicts CD45-positive erythroblasts.

**Figure 6 cells-12-02810-f006:**
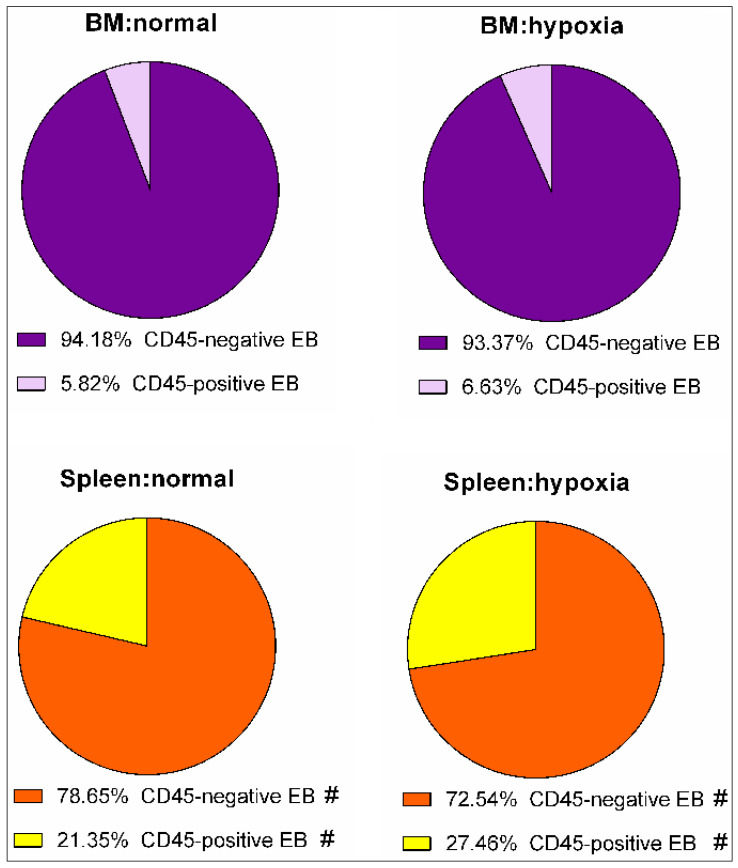
The content (relative number) of CD45-positive and CD45-negative subpopulations among erythroid cells of the spleen and bone marrow in mice under normal and hypoxia conditions. #—differences are statistically significant between the content of phenotypically identical cells in the spleen and bone marrow under the same exposure (*p* < 0.05).

**Figure 7 cells-12-02810-f007:**
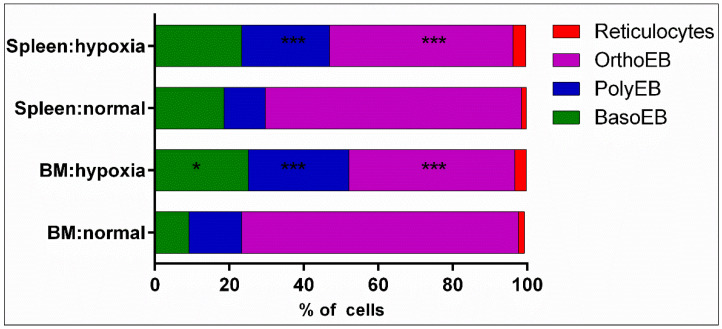
Distribution of stages of terminal differentiation of erythroid cells in the bone marrow and spleen in normal and hypoxia conditions. ***—the differences are statistically significant as compared to the norm (*p* < 0.001); *—the differences are statistically significant as compared to the norm (*p* < 0.05).

**Figure 8 cells-12-02810-f008:**
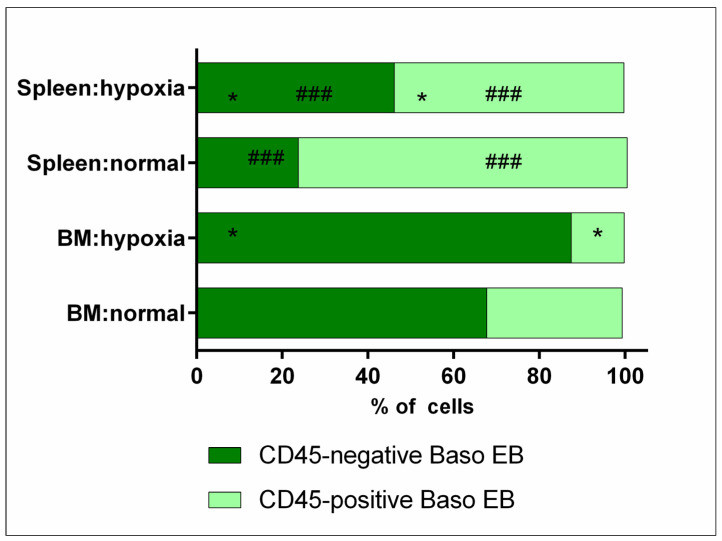
Distribution of CD45-positive and CD45-negative subpopulations among basophilic erythroid cells in bone marrow and spleen under normal and hypoxia condition *—differences are statistically significant as compared to the norm (*p* < 0.05) ###—differences are statistically significant as compared to the norm (*p* < 0.001).

**Figure 9 cells-12-02810-f009:**
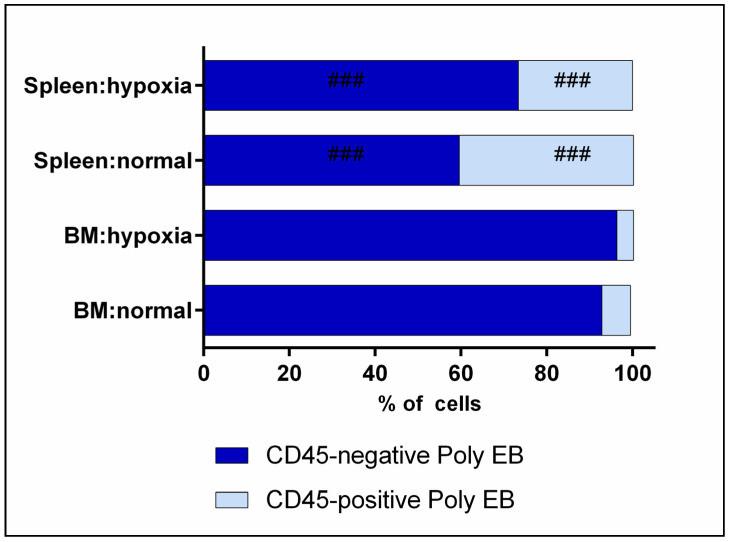
Distribution of CD45-positive and CD45-negative subpopulations among polychromatophilic erythroid cells in bone marrow and spleen under normal and hypoxia conditions. ###—the differences are statistically significant compared to bone marrow cells under the same conditions (*p* < 0.001).

**Figure 10 cells-12-02810-f010:**
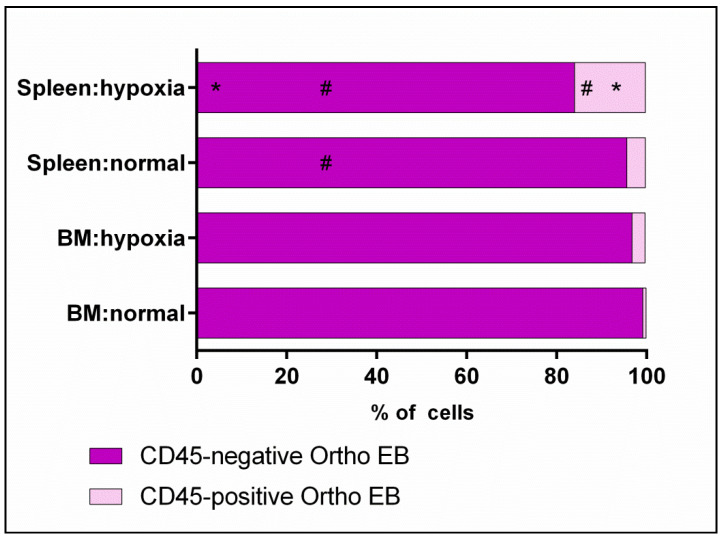
Distribution of CD45-positive and CD45-negative subpopulations among orthochromatophilic erythroid cells in bone marrow and spleen under normal and hypoxia conditions. *—the differences are statistically significant as compared to the norm (*p* < 0.05); #—the differences are statistically significant as compared to bone marrow cells under the same conditions (*p* < 0.05).

**Figure 11 cells-12-02810-f011:**
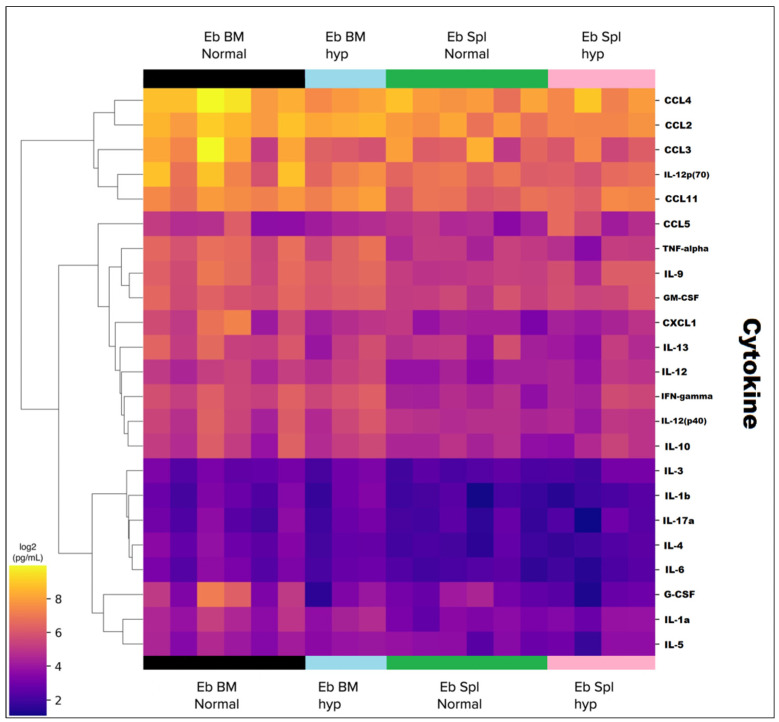
Heat map of the cytokines secreted by CD71-enriched erythroid cells. Eb stands for CD71-enriched erythroid cells, Bm stands for bone marrow, Spl stands for spleen, N stands for normal, and hyp02 stands for acute hypoxia.

**Figure 12 cells-12-02810-f012:**
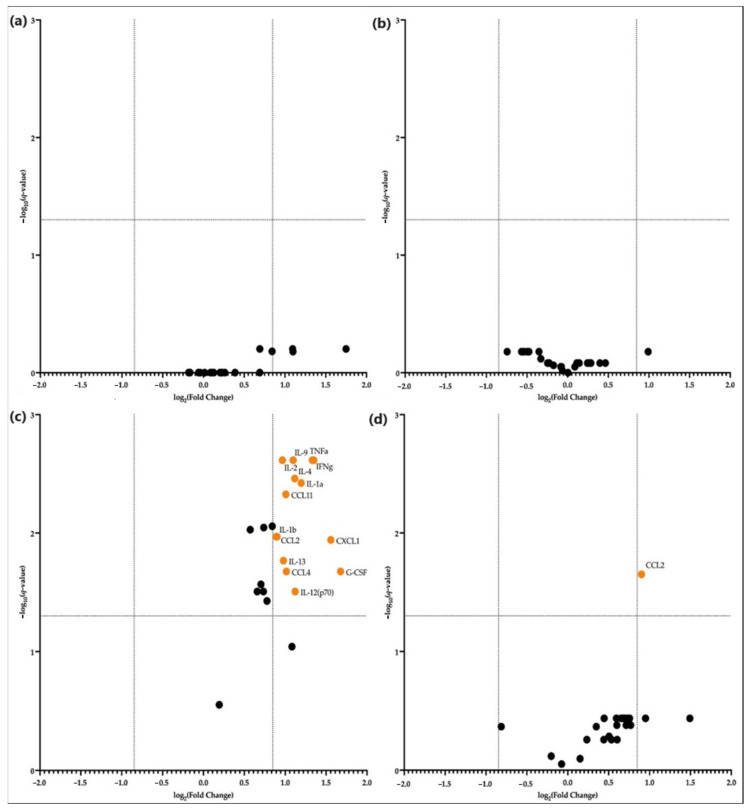
Differentially Secreted Cytokines’ Volcano plot of (**a**) normal CD71+ erythroid cells from the bone marrow vs. CD71+ erythroid cells from the bone marrow after hypoxia; (**b**) normal CD71+ erythroid cells from the spleen versus vs. CD71+ erythroid cells from the spleen after hypoxia; (**c**) normal CD71+ erythroid cells from the bone marrow vs. CD71+ erythroid cells from the spleen; (**d**) hypoxia CD71+ erythroid cells from the bone marrow vs. hypoxia CD71+ erythroid cells from the spleen. Black circles indicate the presence of cytokines for which differential changes are not shown.

**Figure 13 cells-12-02810-f013:**
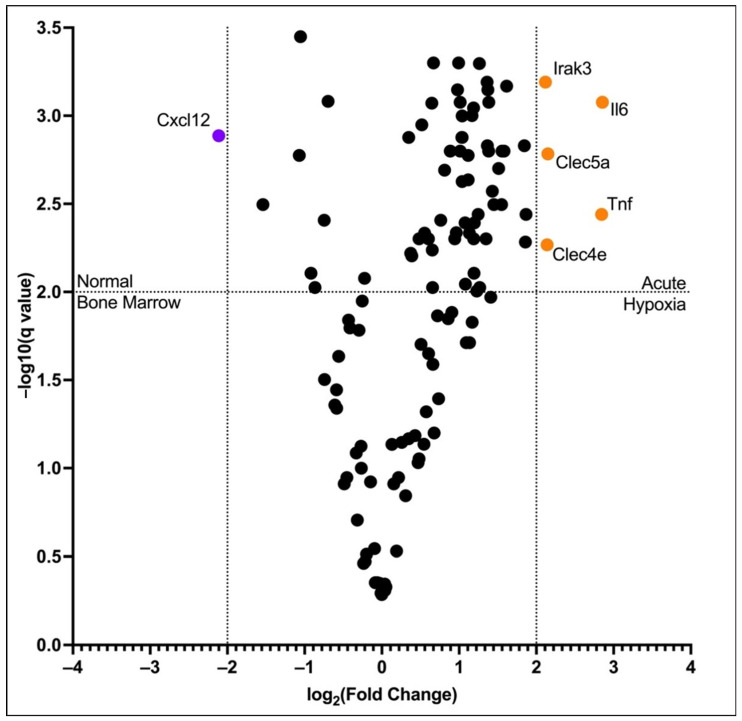
Volcano plot of the differentially expressed genes: we considered *q*-values < 0.01 and log2(FC) > 2.0 or log2(FC) < −2.0 significant; genes in purple were enriched in the CECS, and genes in orange were enriched in the Ter-119-selected erythroid cells. Black circles indicate the presence of genes for which differential changes are not shown.

**Figure 14 cells-12-02810-f014:**
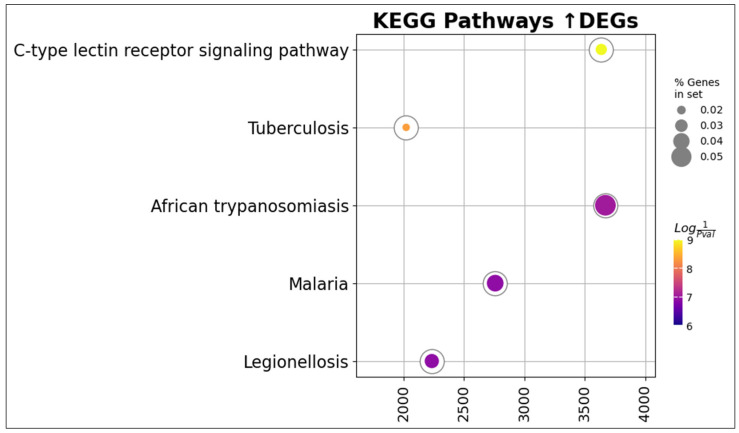
Gene set enrichment analysis of the genes up-regulated (↑) in murine erythroid cells after acute hypoxia.

**Table 1 cells-12-02810-t001:** The content (relative number) of basophilic, polychromatophilic, orthochromatophilic erythroid cells, and reticulocytes in the bone marrow and spleen in normal and hypoxia conditions.

	Baso EB	Poly EB	Ortho EB	Ret EB
BM:normal	9.14 ± 1.73	14.12 ± 0.645	74.38 ± 2.06	1.61 ± 0.75
BM:hypoxia	25.09 ± 6.75 *	27.07 ± 3.54 ***	44.51 ± 5.66 ***	3.10 ± 0.87
Spleen:normal	18.5 ± 4.19	11.22 ± 1.57	68.72 ± 4,58	1.30 ± 0.45
Spleen:hypoxia	23.34 ± 3.94	23.49 ± 1.74 ***	49.38 ± 4.17 ***	3.44 ± 1.00

***—the differences are statistically significant as compared to the norm (*p* < 0.001); *—the differences are statistically significant as compared to the norm (*p* < 0.05).

**Table 2 cells-12-02810-t002:** The content (relative number) of CD45-positive and CD45-negative basophilic erythroid cells in bone marrow and spleen in norm and hypoxia.

	CD45− Baso EB	CD45+ Baso EB
BM:normal	67.68 ± 6.31	31.61 ± 6.09
BM:hypoxia	87.43 ± 2.34 *	12.32 ± 2.28 *
Spleen:normal	23.71 ± 7.38 ###	76.79 ± 7.38 ###
Spleen:hypoxia	46.13 ± 7.27 *,###	53.6 ± 7.18 *,###

*—The differences are statistically significant as compared to the norm (*p* < 0.05); ###—the differences are statistically significant as compared to the norm (*p* < 0.001).

**Table 3 cells-12-02810-t003:** The content (relative number) of CD45-positive and CD45-negative polychromatophilic erythroid cells in bone marrow and spleen in norm and hypoxia conditions.

	CD45– Poly EB	CD45+ Poly EB
BM:normal	92.85 ± 1.92	6.637 ± 1.51
BM:hypoxia	96.35 ± 1.13	3.863 ± 1.09
Spleen:normal	26.56 ± 6.61 ###	73.38 ± 6.61 ###
Spleen:hypoxia	40,57 ± 6.54 ###	59.6 ± 6.52 ###

###—the differences are statistically significant compared to bone marrow cells under the same conditions (*p* < 0.001).

**Table 4 cells-12-02810-t004:** The content (relative number) of CD45-positive and CD45-negative orthochromatophilic erythroid cells in bone marrow and spleen under normal and hypoxia conditions.

	CD45– Ortho EB	CD45+ Ortho EB
BM:normal	99.14 ± 0.31	0.7367 ± 0.31
BM:hypoxia	96.77 ± 1.26	2.859 ± 1.10
Spleen:normal	95.6 ± 1.46 #	4.072 ± 1.51 #
Spleen:hypoxia	83.96 ± 4.40 *,#	15.72 ± 4.32 *

*—differences are statistically significant as compared to the norm (*p* < 0.05); #—the differences are statistically significant as compared to the bone marrow cells under the same conditions (*p* < 0.05).

**Table 5 cells-12-02810-t005:** Gene set enrichment analysis of the genes up-regulated in murine erythroid cells after acute hypoxia.

Term	Overlap	*q*-Value	Score	Genes
C-type lectin receptor signaling pathway	3/112	0.0001	3636.2631	IL6, CLEC4E, TNF
Tuberculosis	3/178	0.0002	2020.3736	IL6, CLEC4E, TNF
African trypanosomiasis	2/39	0.0008	3670.4539	IL6, TNF
Malaria	2/49	0.001	2757.6532	IL6, TNF
Legionellosis	2/58	0.001	2232.8102	IL6, TNF

## Data Availability

The data that support the findings of this study are available from the corresponding author, S.V. Sennikov, upon an email request.

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
