# Peer review of "Phenotypic Alterations in Erythroid Nucleated Cells of Spleen and Bone Marrow in Acute Hypoxia"

_cells, 2023, doi:10.3390/cells12242810_

Round 1

Reviewer 1 Report

Comments and Suggestions for Authors

This manuscript by Nazarov et al. analyzes the impact of acute hypoxia on the immunophenotype of erythroid cells in spleen and bone marrow of mice. They also study the hypoxia induced changes in gene expression and cytokine production. The results provide an immunophenotyic strategy for stratification of erythroblasts and show reduced percentage of nucleated erythroid cells in spleen and bone marrow. Additionally, they demonstrate a hypoxia induced decrease in the fraction of erythroid cells and an alteration in the pattern of erythroblasts and reticulocytes distribution in spleen; there is an increase in reticulocytes, decrease in orthochromatic erythroblasts and accumulation of polychromatic and basophilic erythroblasts. Finally, the data identifies a hypoxia induced upregulation of genes of the C-lectin receptor signaling pathway and downregulation of the cytokine CXCL12. Overall, the study is descriptive, however, provides insights that many be beneficial to researchers in the field.

The following concerns have been identified.

1.     Line 206-208: The Results section states that “Hypoxia in the spleen and bone marrow changes the cellular composition by increasing the content of CD45-positive lymphoid cells and decreasing the content of CD45-negative non-lymphoid cells (Figure 3A)”, however, Figure 3A shows comparison of Lin+/- cells and not CD45+/- cells.

2.     A better explanation of how the HSNE plots were generated and their significance should be provided in the methods and results sections, respectively.

3.     Font size should be increased for labels and legends in scatter plots in Figures 1 and 3, and heatmaps in Figures 5 and 11.

4.     Figure 2: label for the horizontal axis should be changed to “% of nucleated cells”.

Comments on the Quality of English Language

None.

Author Response

Dear reviewer, we thank you for your work on our manuscript. Below we provide answers to your questions and comments.

  1. Line 206-208: The Results section states that “Hypoxia in the spleen and bone marrow changes the cellular composition by increasing the content of CD45-positive lymphoid cells and decreasing the content of CD45-negative non-lymphoid cells (Figure 3A)”, however, Figure 3A shows comparison of Lin+/- cells and not CD45+/- cells.

CD 45 is the predominant linear marker, so we used the terms “linear-positive” and “CD 45-positive” as equivalent. However, for Figure 3A it is really better to use the terms “lineage-positive lymphoid cells” and “lineage-negative non-lymphoid cells”. We have made appropriate changes to the text and captions for the figures. We made similar changes to Figure 4.

2.     A better explanation of how the HSNE plots were generated and their significance should be provided in the methods and results sections, respectively.

We have included a description in the relevant sections.

3.     Font size should be increased for labels and legends in scatter plots in Figures 1 and 3, and heatmaps in Figures 5 and 11.

We have made changes to figures No. 1,3,5 and 11.

4.     Figure 2: label for the horizontal axis should be changed to “% of nucleated cells”.

We have made changes to figure No. 2

Reviewer 2 Report

Comments and Suggestions for Authors

The study is devoted to the investigation of the effect of model of acute high-altitude hypoxia on the phenotypic changes in erythroid nucleated cells of spleen and bone marrow in mice. The authors demonstrated that hypoxia leads to changes in the composition of erythroid cells. In particular, there is a change in orthochromatophilic and polychromatophilic erythroid cells. The level of cytokine production was also assessed. Up-regulation of C-lectin receptor signaling pathway genes was discovered. This study contributes to the understanding of the mechanisms of adaptation of the body to hypoxic conditions. The study was carried out at a high experimental level. The presented results are well substantiated and the manuscript is recommended for publication after Minor revision.

There are the following comments about the study:

-Line 238-240 «The analysis of flow cytometry data demonstrated that the content of CD45-positive erythroid cells in bone marrow was equal to 5.82±1.154% under normal conditions and 6.63±1.673% under hypoxia. In the spleen, under normal conditions, the oxygen content was equal to 21.35±3.763% and under hypoxia, it was equal to 27.46±6.095%.»

«…oxygen content» - typo or unfortunate expression. So, according to Figure 6, this means a CD45-positive subpopulation among erythroid cells of the spleen. Please, correct this. What is the level of p for these values? What is indicated SD or SE? Moreover, it is incorrect to indicate more decimal places in the error than in the value. The same number of decimal places must be specified for the value and the error. The same for the tables. Please, correct this.

-What is known about blood oxygen saturation under such conditions.

-Figure 9. Please, add a description of ### in the caption to Figure 9.

-Abstract – 16-18 «We showed changes in the cellular composition of the bone marrow 16

and spleen during hypoxia, as well as changes in the subpopulation composition of erythroid cells 17

during acute hypoxic exposure in favor of the accumulation of orthochromatophilic erythroid cells 18

ready for rapid enucleation.»

253-255 Under hypoxia, both in the spleen and bone marrow, the content of orthochromatophilic erythroid cells was observed to decrease significantly, while the content of polychromatophilic erythroid cells was observed to increase significantly. (Table 1)

Apparently, there was a typo and the abstract should be corrected to correspond the results.

Comments on the Quality of English Language

Author Response

Dear reviewer, we thank you for your work with our manuscript. Below we provide answers to your questions and comments.

-Line 238-240 «The analysis of flow cytometry data demonstrated that the content of CD45-positive erythroid cells in bone marrow was equal to 5.82±1.154% under normal conditions and 6.63±1.673% under hypoxia. In the spleen, under normal conditions, the oxygen content was equal to 21.35±3.763% and under hypoxia, it was equal to 27.46±6.095%.»

«…oxygen content» - typo or unfortunate expression. So, according to Figure 6, this means a CD45-positive subpopulation among erythroid cells of the spleen. Please, correct this. What is the level of p for these values? What is indicated SD or SE? Moreover, it is incorrect to indicate more decimal places in the error than in the value. The same number of decimal places must be specified for the value and the error. The same for the tables. Please, correct this.

Yes, this is a really unfortunate expression; there was an error when writing the sentence. We are talking about the level of CD 45-positive cells, not the oxygen content. For these values, the p level exceeded 0.05, so there were no significant differences between CD45-positive cells in the bone marrow under normal and hypoxic conditions, as well as in the spleen under normal and hypoxic conditions. Data are presented as the mean and standard error of the mean (mean±SE). I have corrected the number of decimal places in the text in accordance with rounding rules.

-What is known about blood oxygen saturation under such conditions.

In our model, we created hypoxic conditions by reducing atmospheric air pressure. We calculated that this pressure corresponds to a rise to a height of about 4500 meters above sea level. Blood oxygen saturation (saturation) in such conditions becomes below 90%.

-Figure 9. Please, add a description of ### in the caption to Figure 9.

I have added the required designation.

-Abstract – 16-18 «We showed changes in the cellular composition of the bone marrow 16 and spleen during hypoxia, as well as changes in the subpopulation composition of erythroid cells 17 during acute hypoxic exposure in favor of the accumulation of orthochromatophilic erythroid cells 18 ready for rapid enucleation.»

253-255 Under hypoxia, both in the spleen and bone marrow, the content of orthochromatophilic erythroid cells was observed to decrease significantly, while the content of polychromatophilic erythroid cells was observed to increase significantly. (Table 1)

Apparently, there was a typo and the abstract should be corrected to correspond the results.

Yes, there really was an error and we fixed it.